# Dynamic Mechanical Analysis Investigations of PLA-Based Renewable Materials: How Are They Useful?

**DOI:** 10.3390/ma13225302

**Published:** 2020-11-23

**Authors:** Mariana Cristea, Daniela Ionita, Manuela Maria Iftime

**Affiliations:** “Petru Poni” Institute of Macromolecular Chemistry, Aleea Grigore Ghica Voda 41A, 700487 Iasi, Romania; dgheorghiu@icmpp.ro (D.I.); ciobanum@icmpp.ro (M.M.I.)

**Keywords:** dynamic mechanical analysis, poly(lactic acid), polylactide

## Abstract

Interest in renewable polymers increased exponentially in the last decade and in this context poly(lactic acid) (PLA) became the leader mainly for practical reasons. Nevertheless, it is outstanding also from a scientific point of view, because its thermal and morphological properties are offering challenging new insights. With regard to dynamic mechanical analysis (DMA), PLA does not have the classical behavior of a thermoplastic polymer. Often, overlapping events (enthalpic relaxation, glass transition and crystallization) that occur as the temperature increases make the DMA result of a PLA look inexplicable even for polymer scientists. This review offers a perspective of the main phenomena that can be revealed in a DMA experiment and systematizes the information that can be obtained for every region (glassy, glass transition, rubbery, cold-crystallization and melting). Also, some unusual patterns registered in some cases will be commented upon. The review intends to offer indices that one should pay attention to in the interpretation of a DMA experiment, even if the investigator has only basic skills with DMA investigations.

## 1. Introduction

In the world of polymers the pursuit of new materials, intended to solve one or another issue in a specific domain, is analyzed from the point of view of both environmental and economic sustainability. In this regard there are criteria for sustainability connected with the influence on the health of any form of life, with the waste that accumulates in the environment or with the greenhouse effect. Moreover, the costs of all processes have a conclusive role. Despite various views and counterviews related to the sustainable materials, the interest in this field is growing and the market evolves accordingly [1,2,3]. Poly(lactic acid) (PLA) has made a large contribution in the area of bio-polymers: it is biobased, biodegradable/compostable and biocompatible [4,5,6,7]. The starting compound (lactic acid or lactide) is the result of the fermentation of renewable resources (crops like corn, rice or sugarcane). Moreover, degradation occurs under the action of microorganisms and yields products (CO_2_ and H_2_O) that represent valuable compounds for nature. PLA compostability represents a benefit especially when circumstances are disadvantageous for recycling. Poly(lactic acid)—structurally defined as a linear aliphatic thermoplastic polyester—is considered to belong to the third-generation of biodegradable polymers due to its suitable biodegradable and mechanical properties [8].

Due to its biocompatibility and controlled hydrolysis, the first polylactide products were intended for resorbable medical devices [9]. The same attributes recommend PLA for environmentally degradable polymers [10,11,12].

At room temperature the stiffness and strength of amorphous PLA are comparable to polystyrene. Therefore, PLA represents the basis for materials dedicated to disposable objects (cutleries, cups, various containers and pots) or exploited in the packaging and textile industries. However, despite the convenience of PLA for having a good clarity, its brittleness oriented a great number of investigations to ameliorate its toughness [13]. A good toughness requires a good balance between strength and ductility.

The structural changes of PLA are reflected in the modification of molar mass/molar mass distribution and in the nature of the chemical groups found in the polymer chains. Specific techniques are used for this approach: gel permeation chromatography (GPS), matrix-assisted laser desorption-ionization time-of-flight mass spectrometry (MALDI-TOF-MS), nuclear magnetic resonance (NMR) and Fourier-transform infrared spectroscopy (FTIR). Morphological transformations are followed with the aid of scanning electron microscopy (SEM), transmission electron microscopy (TEM) or atomic force microscopy (AFM). A close and continuous examination of the literature connected with the characterization of PLA and PLA-based polymers revealed that differential scanning calorimetry (DSC) is largely used in various instances, sometimes in association with wide angle X-ray scattering (WAXD). This fact is explained by both scientific and practical reasons, thermal behavior of PLA being expressed in the sequence glass transition–crystallization–melting, both on the heating and on the cooling step.

The glass transition temperature of PLA (T_g_) lies between 50 °C and 70 °C, the morphological features being the main reason for the shift of T_g_ in this interval. As a rule, this region is characterized by long-range coordinated molecular motions and pronounced changes of physical characteristics of polymers (i.e., heat capacity, coefficient of expansion, storage modulus) [14]. In the case of PLA the glass transition comes along with other phenomena (stress relaxation, orientation/crystallization, shrinking) that prompt scientific curiosity, but rise supplementary issues in practice. The cold crystallization that follows the glass transition is largely used as a means to improve the heat distortion temperature (HDT) in order to increase the thermal stability at higher temperatures. The melting region is important for establishing processing conditions or for clarifying details of crystalline forms.

Frequently, dynamic mechanical analysis (DMA) investigations are added to various thermal studies with the declared aim to confirm the DSC results or to complete them. Nevertheless, looking in the PLA literature, there are cases where the DMA results are not exploited enough or some interpretations are incoherent. There are two main reasons for this fact. On the one side, rather unusual patterns are noticed in the variation of the viscoelastic parameters vs. temperature (T) in the case of PLA. An investigator not accustomed to the DMA technique might have difficulties in understanding behaviors that do not resemble standard diagrams, mostly because numerous overlapping and successive phenomena that occur in PLA with increasing temperature. On the other side, the new generation of instruments is updated according to digital technologies and the results are obtained much more easily than before. Accordingly, less time and endeavor is dedicated for interpreting the results.

Since PLA has become the center of attention in the world of polymers from renewable resources, many facets of it have been reviewed: obtaining, properties and characterization [15,16,17,18,19], toughening, crystallization and weathering aspects [13,20,21,22], composites and blends [23,24,25], packaging and biomedical applications [26,27,28]. Therefore, only a short overview of poly(lactic) acid will be included at the beginning, with the features that are utterly needed to make the DMA part understandable even for somebody who is not familiarized with the class of polymers. Afterwards, some theories describing the DMA will be recalled succinctly, as necessary. The main objective of this overview is to present the salient usefulness of DMA in the characterization of poly(lactic acid) and its derivative materials. What are DMA advantages beside DSC and in which situations are the DMA results not reliable? The review will not focus on a specific category of PLA system, but on a peculiar DMA behavior.

## 2. Poly(Lactic Acid) (PLA)—General Outlook

The constituent unit of PLA is lactic acid or 2-hydroxy propionic acid. The presence of chiral carbon atom makes it exists as two enantiomers: L-lactic acid and D-lactic acid. Both forms of lactic acid are able to undergo cyclization, with the formation of dimers: homochiral lactides (L,L- and D,D) or heterochiral lactide (meso-lactides), the last one including two chiral atoms with different configurations [29]. Agricultural crops represent the main source of lactic acid (L or D), as a result of bacterial fermentation of carbohydrates. The racemic mixture is obtained from the synthesis. Accordingly, PLA has the following stereoisomers: poly(L-lactic acid) (PLLA), poly(D-lactic acid) (PDLA) and poly(DL-lactic acid) (PDLLA) (Scheme 1). Various polymerization processes are used for obtaining PLA: polycondensation of lactic acid, ring opening polymerization of lactides and direct methods, azeotropic dehydration or enzymatic polymerization. The first synthesized PLA (Carothers, Du Pont, 1932) had a low molecular weight. By contrast, the ring-opening polymerization of lactide results in high molecular weight PLA [17]. The nomenclature poly(lactic acid)/polylactide is employed interchangeably, with a preference for polylactide in the situation of high molecular weight polymers. The ratio of L- to D- enantiomers is known to impact on the properties of PLA, mainly on the degree of crystallinity. PLA that comprises LLA content close to 100% is highly crystalline, while a ratio LLA/DLA of 1/1 makes the structure to be amorphous. Some publications report the minimum content of LLA required for the development of crystallinity during processing as being higher than 85%. As the content of D isomer is higher, PLA has less crystallinity. Also, PLA develops various crystal forms: α, α’, β, η (stereocomplex) [20,30,31,32].

In this article the abbreviation PLA will be used; however, information about stereoregularity will be mentioned if necessary.

The preponderance of PLA practical applications requires the use of PLA in combination with additives to mitigate its drawbacks: low HDT and small impact strength [13,33,34,35]. The low HDT results mainly from the reduced glass transition temperature of PLA (50 °C to 70 °C) and the reduced crystallization rate during processing. An appropriate crystallinity would confer load-bearing properties to PLA at temperatures higher than T_g_. Also, the use of nucleating agents to support crystallization and fillers to reinforce the PLA will overcome the non-complying HDT [36,37]. The brittleness of original PLA makes its materials vulnerable to a sudden applied load. Therefore, PLA is largely used in combination with plasticizers and toughener polymers [38,39,40,41,42].

## 3. Dynamic Mechanical Analysis—Basics and Rationale of Its Usefulness for PLA

The word of materials started to reshape substantially a century ago once polymers stood out as peculiar chemical compounds. Accordingly, the methods of characterization have been developed in order to measure up their specificities. Their time-dependent properties (mechanical, electrical, optical) are the result of their long-chained structure. Dynamic mechanical analysis was largely used to characterize the viscoelastic properties of polymers as they involve both solid and liquid-type characteristics.

In DMA a sinusoidal load (stress or strain) is applied to a sample and the corresponding response (strain or stress) is recorded.

When a sinusoidal stress is applied, it is described by the equation:(1)$$\sigma =\sigma_{0}\sin\omega t$$
where σ is the stress at time t, σ_0_ is the amplitude of the stress and ω is the angular frequency.

The resulting strain is as well sinusoidally shaped:(2)$$\gamma =\sin\left(\omega t+\delta \right)$$
where γ is the strain at the time t, γ_0_ is the amplitude of the strain and δ is the phase difference between the stress signal and the strain signal.

The phase lag δ is 0 for ideal elastic solids that respect the Hooke’s law. The phase lag δ is 90° for perfect viscous solids, according to the Newton’s law. The viscoelastic materials have 0°< δ < 90°.

Accordingly, a complex modulus is defined that includes both the elastic part (E’) and the viscous part (E’’):(3)$$E^{*}=E^{\prime }+iE^{\prime \prime }$$
(4)$$E^{\prime }=E^{*}\cos\delta \mathrm{and}E^{\prime \prime }=E^{*}\sin\delta$$
where E’ is the elastic (storage) modulus, connected with the rigidity of the sample, and E’’ is the viscous (loss) modulus, correlated with the flowing properties.

The loss factor (damping) is defined as the ratio between the viscous modulus and the elastic modulus:(5)$$\tan\delta =\frac{E^{\prime \prime }}{E^{\prime }}$$

In terms of energies, the loss factor represents the ratio between the dissipated energy and the elastically stored energy.

The scientific literature includes some featured references that treat thoroughly many facets of DMA [43,44,45,46,47,48,49]. Despite the fact that the majority of experiments are performed isochronally, with the variation of temperature, it is important to understand the method as being part of rheology. Oscillatory tests in rheology are known as dynamic mechanical analysis [50]. The DMA method helps in understanding the solid-state rheological properties of polymers that are expressed by relaxation phenomena.

Figure 1 pictures schematically the result of a single frequency DMA experiment performed on a thermoplastic elastomer. From the very low temperature to high temperatures the behavior of the polymer changes according to its molecular mobility. There are four regions: sub-glass transition (glassy) region, glass transition region, rubbery plateau and flowing region.

In the glassy region the macromolecular chains are frozen in a rigid structure (E’ > 10^9^ Pa). In this zone only local motions of side groups or backbone movements of bending/stretching are possible (γ and β secondary relaxations). With increasing temperature the long-range coordinated molecular motions (α-relaxation) are activated. It is well-known that α-relaxation is associated with the glass transition temperature. This zone is clearly marked by a significant decrease of E’ (usually, three orders of magnitude for an amorphous polymer) and well-defined tan δ and E’’ peaks on the respective curves. These peaks do not coincide and their values are reported as T_g_ values. The experimental conditions (heating rate, frequency or loading type—tension, bending, compression, shear) influences meaningfully the results [51,52]. The characteristics of the rubbery plateau (the E’ value, the length or the existence of other phenomena like melting or cold crystallization) depend on the molecular weight and on the morphology of the polymer. Usually, in the flowing region the mobility of the macromolecular chains is so high that they slip one past another. At this point the polymeric material is no longer a load-bearing sample and the experiment cannot continue further. It is evident that the flowing region is not present in the case of thermosets.

As a matter of fact, the representation outlined in Figure 1 denotes the pattern of a well-behaved polymer. The sources and processing of familiar classes of polymers are diverse; also, the structures of polymers are much more complex as the result of sophisticated synthesis. Therefore, a real result of the DMA experiment can become a puzzle for neophytes of the method, even for the determination of the T_g_. Not every peak registered on the variation of tan δ and E’’ vs. T or downward trend of E’ vs. T can be associated with a relaxation. Overlapping events can result in intricate patterns of the viscoelastic characteristics vs. temperature. Materials based on PLA could be in one of these cases, especially when PLA is modified or used in combination with additives to smooth out its drawbacks.

It is meaningful to follow the trends of the elastic modulus E’ with increasing temperature, as long as it is related to the load-bearing properties of the PLA material. Many publications reported the use of HDT, despite the fact that it represents only a single temperature at which the E’ modulus diminishes to a specific value in standardized conditions. In a series of well-documented articles, M. Sepe advocated the use of DMA instead of HDT [53], because DMA gives the entire description of the load-bearing properties in a large temperature range, in the same time interval.

Among the thermal analysis techniques that were used to characterize the thermal properties of PLA-based polymers, DSC is the most convenient. However, there are situations where, for instance, the filler used as reinforcement constrains the PLA chains insomuch, as the heat capacity jump becomes undetectable in DSC. By contrast, the well-quoted DMA tan δ peak is very reliable in identifying the glass transition region. Even so, the shape of the tan δ peak of PLA-based compounds is often distorted because of overlapping phenomena that occur in the glass transition region. These general arguments are a good starting point for our demarche to elaborate an overview about the meanings and limitations of using DMA in the investigations of PLA-based materials.

## 4. Dynamic Mechanical Analysis—The Unique Viscoelastic Behavior of PLA

The pioneer works related to the viscoelastic behavior of polylactide date from the beginning of the ‘90s and reported similar results [54,55,56]. The nature of PLA explains the small differences between them (Table 1), mainly caused by different molecular weights or degrees of crystallinity of PLAs under study. Table 1 summarizes the main characteristics of the starting PLA and DMA experiment, as they were reported from the selected studies cited in the references’ part.

A schematic representation of the E’ modulus vs. T for an amorphous PLA adapted after these publications is presented in Figure 2.

A high E’ modulus close to 5 × 10^9^ Pa defines these PLA samples in the glassy region (A). A sharp descent of E’ takes place around 60 °C, where the glass transition region (B) is centered. On the rubbery plateau (C), that can be shorter or even absent for low-molecular weight PLA, the PLA chains have sufficient mobility to develop crystalline regions. A sudden increase of E’ modulus indicates the cold crystallization (D). The second rubbery plateau (E) evidences a stiffer PLA as a result of the confinement effect of crystalline arrays exerted on the diminished amorphous phase. At the melting point, over 140 °C, the E’ modulus decreases abruptly (F). The E’ value becomes too low to allow the DMA experiment to be continued or even kept isothermally at the melting temperature.

### 4.1. The Glassy Region

Studies which focused on the secondary relaxations of PLA are rather scarce. There are a few specific examples of a secondary relaxation, albeit the single evidence is the slow descent of E’ modulus with increasing temperature and a large tan δ peak. Starkweather et al. mentioned a secondary relaxation at −50 °C for a polylactide (~95% L-isomer) [56]. Courgneau et al. also reported a secondary relaxation (β-relaxation) at −83 °C for a poly(lactic acid) (~92% L-isomer) [58]. Celli and Scandola pertinently justified the low discernible secondary relaxations by the brittleness of glassy PLA [55].

The brittleness can be an impediment also from a practical point of view, because the DMA experiment requires to fix in a clamp a polymer sample having a well-defined shape. Sometimes it is impossible to load correctly this kind of sample at room temperature; therefore, even more difficult would be to perform the DMA measurements at negative temperatures.

Since the glass transition temperature of PLA is close to 60 °C, this kind of polymeric material is in the glassy state at room temperature. Various fillers (glass fiber, nanocellulose, wood-based fibres BKSW), modified polyhedral oligomeric silsesquioxane—POSS, reduced graphene oxide—rGO, nanorod-shaped organic-inorganic hybrid material, talc) are used in combination with PLA with the declared aim to improve the load-bearing properties [59,60,61,62,63,64,65,66,67,68]. In the glassy region a non-logarithmic representation of E’ vs. T helps always to track better the change of E’ modulus and to compare visually composites with variable contents of filler. In Figure 3a and Figure 4a the reinforcement effect of wood-based fibers and talc in the glassy region is comparatively emphasized, in a logarithmic and non-logarithmic E’ vs. T plot, for two commercial PLAs [63,67]. The non-logarithmic plot displays better the differences between the samples in the glassy region.

One would expect that increased filler content strengthens the PLA composite and results in higher E’ values. Apparently, E’ modulus is not determined solely by the content of the filler. Ferreira and Andrade [65] explain this behavior by the propensity of graphene oxide to reduce the crystallinity of PLA. In other instance, when POSS [64] and organic-inorganic hybrid material (composed of aluminum hydroxide—ATH and benzenephosphinic acid—BPA) [66] were used as fillers, the reinforcing at low quantities of filler is more effective than at higher content. An increase in the filler content does not result in a coherent augmentation of the elastic modulus E’. The composite with 20 % ATH-BPA (PBA3) has a lower E’ than the composite with 10% filler (PBA1) (Figure 5a).

The same trend was reported for PLA-POSS composites (1, 2.5, 5, 10 and 20 wt% POSS content). The composite with 20 wt% POSS has even a lower E’ modulus than the neat PLA [64]. A reason for this trend could be the fact that additional filler either spaces out the PLA chains because the lack of interaction with them, or agglomerates and acts as a weak point in the material.

DMA performed after experiments that simulate the degradation of a commercial polylactide (3.8 % meso-lactide content) in soil shows the decrease of the E’ modulus in the glassy region with the time spent in soil (Figure 6a) [68]. The formation of shorter chains is the dominating factor behind the weakening of the PLA system.

### 4.2. The Glass Transition Region

The DMA technique is often the method of choice for the determination of T_g_ when the signal registered by DSC is too faint. No matter what value of temperature is considered to be the T_g_ (the onset drop of E’ or peaks of E’’ or tan δ), PLAs with different crystallinity or PLAs that include additives (fillers, nucleating agents or plasticizers) show very small variation of this temperature with the perturbing factor. Even a glance at the tables that encompass the main parameters of DMA or DSC results reveals this aspect [59,63,67,69,80]. The reason for this may be the concurrent events that may occur in opposite directions during the glass transition region and the particular morphology of PLA. Some of them will be described below.

Picciochi et al. [32] investigated by DMA the glass transition region of amorphous PLA, cold-crystallized at various temperatures to get different crystallinity. DMA experiments revealed that the T_g_ (tan δ peak) shifted towards lower temperatures as the crystallinity increases. The reverse trend would be expected, since increased crystallinity means constrained movements of the polymer chains. The authors explained this apparent unusual trend in terms of cooperatively rearranging regions (CCR) that suppose the existence of dual amorphous phases having dissimilar mobilities: mobile amorphous phase (MAF) and rigid amorphous phase (RAF) [81,82]. The increase of the RAF layer thickness will induce the decrease of T_g_ since it has some degrees of freedoms as opposed to the crystalline phase.

The hump that is registered on the E’ curve just before the onset point is associated with enthalpic relaxation [44,70,83], a phenomenon that is better accentuated in a DSC experiment as an endothermic peak overlapping the glass transition [54]. Figure 7 exhibits comparatively the overlapping of the glass transition with enthalpic relaxation in a DMA and DSC experiment performed on poly(lactide-co-glycolide) (PLGA) non-woven fabrics. Internal stresses are accumulated during PLA processing because the polymer chains are at least partially extended and then cooled rapidly. There is not enough time for the macromolecular chains to go back to thermodynamic equilibrium (aging) [84]. In the glassy state the chains are frozen in a non-equilibrium state. As the temperature is going up, the onset of the glass transition triggers the refolding of polymer chains because their mobility raises. As a matter of fact this increased entropy is actually the driving force for shrinkage [85].

What cannot be overlooked when dealing with physical aging is the thermal shrinking during the glass transition [86]. This affects significantly the viscoelastic behavior in this region, and the polymer becomes stiffer. Pluta and Galeski reported that the T_g_ determined in a DMA experiment (tan δ peak) performed on poly(L/D,L-lactide), deformed to different ratios, can be higher than in reality [71]. Experiments of thermomechanical analysis (TMA) emphasized the dimensional change of PLA samples as the temperature increased [63,87,88]. Nevertheless, even a DMA instrument registers the position of the drive shaft in real time, making possible to follow the evolution of the sample length during a tension experiment. Shrinking of non-woven PLA fabrics (Figure 8) [69] and of L-lactide/glycolide/trimethylene carbonate terpolymers [89] were recorded during DMA investigations in the glass transition zone.

All these changes are reflected as well in the plot of E’’ as a function of temperature. It exhibits an unusual sharp peak when shrinking is present during the glass transition, as is shown in Figure 6b [68]. This shape is consistent with a sudden cease of the coordinated molecular motions—the main feature of the glass transition—on account of an instantaneous increase of rigidity. Investigators generally avoid displaying the pattern E’’ vs. T, most likely because of the altered shape that induces confusion. There are some reports that use the E’’ representation [42,80], but still the description tan δ vs. T is by far the favorite one because the peak associated with the T_g_ is best-known. Even so, the comments are mostly focused on the tan δ peak and many details of the tan δ vs. T plot are overlooked. The tan δ vs. T plot should be exploited to the fullest.

As the crystalline phase becomes more important, the PLA sample gains dimensional stability and the shrinking lessens. This was assessed for PLA/BKSW [63] biocomposite or PLA/talc composites [67]. The stiff framework within the composites arrests the segmental motion of the polymer and this is reflected in the decreasing of tan δ amplitude as the filler quantity augments (Figure 3b and Figure 4b). Because the rigidity of the polymer is reflected in the E’ value, a higher E’ will result in a lower E’’/E’ ratio (tan δ).

Analogously, it is expected that the amplitude of tan δ increases when PLA is used in combination with plasticizers, since the toughness of the mixture is improved as compared to original PLA. An increased toughness is reflected in a higher E’’ modulus at the expense of rigidity (E’ modulus); therefore, the ratio E’’/E’ (tan δ) augments. In the blends composed of PLA/epoxidized soybean oil [72] and PLA/acetyl tributyl citrate [58] the effect of plasticizers is reflected in reduced E’ values in the glassy region and lower T_g_ values. However, the dynamics of events during the glass transition make the tan δ peaks decrease as more plasticizers is added, an evidence for extra-stiffness. This trend is opposite to the above rationale. Stiffening phenomena caused by shrinking might be at the origin of these behaviors.

Moreover, the reinforcement effect reflected in increased E’ modulus in the glassy region in PLA composites reinforced with glass fiber [59] and nanocellulose [61] is not displayed as a decrease of the tan δ height because of reduced mobility. By contrast, high quantities of nanocellulose caused a major increase of tan δ. The authors do not explain this, but the profile may be caused by similar factors as those mentioned by Cao et al. for the PLA/hybrid material represented in Figure 5b [66]. Namely, a small quantity of ATH-BPA filler (PBA1, 10%) induces the decrease of tan δ amplitude as compared to PLA. However, more filler (PBA3, 20%) makes the tan δ height increase up to a value close to that of original PLA. As it turns out, the additional filler does not serve as a fortifier. Refolding of polymer chains as a result of enthalpic relaxation could make the filler to be unevenly distributed throughout the network and zones with higher mobility are created.

As it was mentioned in the introductory part, the glass transition region becomes even more complex, since also orientation/crystallization events could be triggered. The particularity of scanning temperature DMA experiments, that imply the simultaneous action of temperature and force, make them possible. Even if the stress (strain) applied to PLA during the DMA experiment is small, stress (strain)-induced ordering phenomena were described in the glass transition region. Some authors mentioned the occurrence of an interplay between the polymer chain mobility (refolding) and stress (strain) induced crystallization that decides the configurations of polymer chains [90,91,92]. Also, there are attempts to describe the nature of the ordered domains either as an imperfectly order state (nematic-like order) [72] or as a short range ordered structure (mesomorphic phase) [91,92]. These events take place independent of the cold-crystallization, but the oriented domains that can be formed during the glass transition may have consequences for it.

### 4.3. The Cold-Crystallization Region

The onset of the cold crystallization is marked in DMA by an instant and abrupt increase of E’ modulus. As has been stated above, the cold crystallization process may be triggered even during the glass transition. This behavior is displayed as a sudden increase of E’ that comes after the glass transition. When there is no rubbery plateau after the glass transition, the variation E’ vs. T has a V-shape. Plasticized PLAs exhibit this kind behavior because the flexibility of the PLA chains is notably raised. There are data that follow the V-shape DMA pattern. They were reported for PLA plasticized with epoxidized soybean oil [72], adipates [73], dioctyl phthalate (DOP) [74], thermoplastic starch [75], oligomeric malonate esteramides [76] and tributyl citrate oligomers [93]. Figure 9a,b displays the viscoelastic behavior of PLA plasticized with DOP.

A change of slope is seen on the E’ vs. T curve (Figure 9a) even before turning the decreasing trend of E’ into an increasing trend. This can be an indication for the kick-off of cold crystallization even during the glass transition. The onset of the cold crystallization, estimated as the E’ upturn point, occurs at a progressively smaller temperature as the quantity of DOP is higher. Again, the decreasing trend of tan δ peak height with increasing plasticizer content is not consistent with the typical pattern of tan δ when toughness is improved. Also, it is worth noticing the sharp shape of E’’ peak (Figure 9b). All these are attributed to the peculiarities of PLA glass transition detailed in Section 4.2.

Composites of PLA/talc [67], mentioned and discussed already in Section 4.1., feature the cold crystallization firstly as a more evident change of slope of E’ vs. T curve, for low talc content (Figure 4a—neat PLA and PLAT1). This is reflected also as a second smaller peak on tan δ curve (Figure 4b).

However, most often the glass transition peak and crystallization peak/shoulder are not separated. They are reunited in an askew peak, with the downward side less abrupt (Figure 3b and Figure 5b) as was reported by Espinach et al. and Yazdaninia et al. in the examples mentioned in Section 4.1 [63,64]. The E’ first rubbery plateau (zone C, Figure 2) is clearly emphasized in these two systems.

The occurrence of cold crystallization depends on the stereoregularity. Hu et al. investigated PLA/poly(ethylene glycol) (PEG) blends that contains PLA with low and high stereogularity [77,78].

The large drop of E’ associated with the glass transition is not followed by cold crystallization for PLA/PEG blends with low stereoregularity (Figure 10a). Cold crystallization occurs only in the mixture based on PLA with high stereoregularity (Figure 10b).

It is interesting to observe the E’ vs. T plot represented in Figure 10b. Seemingly, the behavior of the 70/30 PLA/PEG blend could be interpreted as an exception, since the increase of E’ associated with the cold crystallization appears at the highest temperature. Actually, the slope change that overlaps/follows the glass transition is the real onset of crystallization that occurs slowly at the beginning. The other two PLA/PEG mixtures (90/10 and 80/20) display the V-shape variation of E’ vs. T curves in the regions glass transition-cold crystallization.

Badia et al. presented the behavior of mechanically recycled PLA (Figure 11) [57]. The length of the rubbery plateau decreases as the number of recovery cycles is higher. Chain cleavage occurs during mechanical recycling and the shorter chains crystallize at lower temperature. The same evidence resulted from investigation of PLA samples submitted to different degradation environments [68] (Figure 6). The rubbery plateau shortens as the degradation time is longer.

Yang et al. demonstrated how the crosslinking degree influences the viscoelastic behavior on PLA crosslinked by chemical treatment, with triallyl isocyanurate (TAIC) and dicumyl peroxide (DCP): the higher the quantity of the crosslinking agent, the less intense is the tendency to crystallize and the rubbery plateau is longer [79].

The heating rate is a critical experimental condition when working with DMA mainly because the method uses quite big samples. This is why small heating rates are required to avoid thermal heterogeneities and high thermal lag between the sample and the thermocouple. When physical or chemical processes take place during the DMA experiment, the heating rate becomes more important [51]. Tabi et al. demonstrated how the rise of the heating rate changes the DMA behavior of PLA [35]. With increasing heating rate the cold crystallization becomes less important (Figure 12). Actually, the polymer does not have enough time to crystallize at very high heating rates (15 °C/min and 20 °C/min) and its DMA behavior looks like that of a standard thermoplastic elastomer.

After the cold crystallization, the E’ modulus levels off on a second rubbery plateau. No important events occur until melting. It is possible that crystallization may continue at a very slow rate, but it cannot result in notable transformations in the time span of the experiment [54]. Nevertheless, Tabi et al. [31] noticed the occurrence of recrystallization from the less ordered α’-form into the more ordered α-form, as a small increase of E’ modulus just before melting.

### 4.4. The Melting Region

An essential requirement for performing a DMA experiment is the load-bearing properties of the samples. When the sample softens, as happened once the melting starts, the DMA experiment cannot be continued and it is not possible to obtain any kind of information in this area. Nevertheless, Sikorska et al. noticed that a kind of contraction takes place at the beginning of melting in the case of PLA-based non-woven fabrics [69]. This contraction may be associated with a welding process, described before by Ljungberg et al. [94]. At the melting temperature the polymer chains are able to diffuse across the interface and join by welding. Also, this phenomenon can be attributed to the premelting crystallization [95] clearly evidenced by DSC as a faint exotherm peak just before the endotherm peak associated with melting.

## 5. Conclusions

A survey of literature oriented to the use of the DMA technique in the study of polymers reveals that the tool is too often used only for the determination of the glass transition temperature. This is obtained mostly from tan δ peak of the plot tan δ vs. T, which resulted from a scanning temperature experiment, performed at one frequency. Nevertheless, an accurate understanding of the viscoelastic behavior of polymers requires the comparative examination of all the viscoelastic characteristics: elastic modulus E’, viscous modulus E’’ and loss factor tan δ. Such an approach is even mandatory in the case of PLA and PLA-based polymers that involve a whole hierarchy of phenomena which are triggered as the molecular mobility is changing from low temperature to high temperature. These aspects are summarized below.
In the *glassy region* (T < T_g_) the secondary relaxations are incidentally mentioned at −50 °C or lower. The β-relaxation was evidenced as a faint drop of E’ modulus or a shallow tan δ peak. Because of the brittleness of PLA, the DMA device is not able to perform reliable experiments at negative temperatures on samples that have a propensity toward cracking.When dealing with PLA, the *glassy region* means also room temperature condition. Therefore, the DMA investigations allow the determination of the elastic modulus E’ under usual working conditions.The effects that are noticed in the *glassy region* during composition-dependent studies are reported also for other classes of polymer. The particularity comes from the semicrystalline character of PLA. The processing conditions, the nature and the content of stereoisomers determine decisively the morphology of PLA in terms of crystallinity. Crystalline content can be tuned during the processing stage, inducing an envisaged change of the E’ modulus.By far the most challenging zone is *the glass transition region*. The chain mobility may be influenced, besides the temperature, by the history of the polymer (aging phenomena) and the applied stress/strain that can induce orientation effects.Typically, the glass transition temperature is considered to be the onset of E’ drop or the peaks of E’’ or tan δ. These indicators are often ambiguous in the situation of PLA because of overlapping phenomena that happen during the glass transition. The synergism of enthalpic relaxation, coordinated molecular movements and orientation/crystallization phenomena makes the determination of T_g_ by DMA fraught with difficulties.The E’ onset is often hidden by a hump that could appear just at the beginning of the glass transition on the E’ vs. T plot, because of enthalpic relaxation. As a result, contraction of the samples is obvious when the DMA experiment is performed under tension loading.The peaks of E’’ and tan δ for PLA are also deformed as compared to those of a well-behaved polymer that records during the glass transition only the coordinated movements of chain segments.The E’’ peak appears very sharp. This E’’ shape accounts for an instantaneous break of mobility growth due to refolding of polymer chains (shrinking).The tan δ peak is at least bimodal, its descending side is less abrupt, larger than the ascending side, and it may span partially the first rubbery plateau and the cold crystallization region. This is consistent with few underlying processes. Therefore, under the simultaneous action of temperature and force, orientation/crystallization phenomena are triggered even during the glass transition.The increase of crystalline content does not entail necessarily the increase of the glass transition temperature. From a certain level of the crystalline content upward, the values of the glass transition temperature decrease with the crystalline content. These results should be discussed in terms of cooperatively rearranging regions (rigid amorphous phase and mobile amorphous phase).The increased toughness that is obtained by adding a plasticizer is reflected in a lower T_g_, but very often the height of tan δ peak monitored during the glass transition region decreases with the toughener quantity. Similarly, a reinforcement agent augments the E’ modulus. However, an opposite effect is reflected in the height of tan δ, i.e., it may increase as more reinforcement is included in the DMA. These patterns are consistent with the effects of already mentioned overlapping phenomena happening during the glass transition (enthalpic relaxation, shrinking, orientation/crystallization).The length of *the first rubbery plateau* depends on the PLA molecular weight. It can be considered as a gauge for the PLA level degradation in decomposition studies.In the presence of efficient plasticizers the first rubbery plateau is absent because the cold crystallization begins during or immediately the glass transition region.The *cold crystallization* is evidenced by a sudden increase of E’ modulus. When it follows the glass transition region (the first rubbery plateau is absent), the E’ vs. T plot has a V-shape.There are instances where the cold-crystallization is not encompassed by the extended tan δ descending side. It can appear as a separate, smaller, frequency-independent peak.The E’ value of the second rubbery plateau is lower than that of the glassy region; however it is stable until the abrupt decrease at melting.With regard to the heating rate, it is evident that its value is meaningful firstly for the point of view of DMA investigation accuracy. Then, the kinetic events that might take place as the temperature is raised require time for completion. Heating rates higher than 2–3 °C/min are not adequate for fulfillment of both conditions.

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
