# Peer review of "Dynamic Mechanical Analysis Investigations of PLA-Based Renewable Materials: How Are They Useful?"

_materials, 2020, doi:10.3390/ma13225302_

Round 1

Reviewer 1 Report

Overall, the manuscript provides important new insights into the use of a widely used polymer. The reference journal articles are fairly recent, and the images compiled by the authors offer a comprehensive review of existing methods to analyze DMA data. 

My only suggestion is that since in the abstract, the authors set out to make DMA data interpretation fairly simple even for a beginner/non-expert, it would be great if the authors could summarize their findings in the form of bullet points in their conclusion. This could serve as a guide for non-experts. 

Reviewer 2 Report

In general I liked the article, I think this time of paper could be very helpful for other researchers with less experiences with the method. The general introduction should be improved, figure captions should contain more description about the material composition and eventually the discussed feature of a curve should be concretized with a mark in the picture. Sometimes the explanation of the phenomenon could be longer and more detailed.

L32 usually it is not the waste material that is processed

L33-34 not very comprehensible sentence

In general English must be deeply revised and improved. The language is comprehensible but not nice.

L83 class of lactides ?

L117 reduced or high Tg ?

L144 Hooke’s law

The introductory chapter about DNA is much better than the general introduction

Figure 2 the figures should be self-explanatory, so the sections A-F should also be explained in the caption.

L226  rather missing?

L252 grapheme?

Figure 4 the percentage of talc should be specified

L267 I cannot see the described changes

L290 The addition of a figure (DSC) could be helpful.

Figure 7 Please, explain NWM1-5. In general It could be helpful sometimes to add an arrow or other mark pointing at the particular spot of question on the picture.

Figure 10 Explain symbols in the figure.

Reviewer 3 Report

The article is very well written, describing in details the thermo-mechanical behavior of PLA and bringing good insights into the field. Several statements are indeed very interesting.

The number of Figures is sufficient, but they need a quality improvement.

I would disagree only with one statement in the Introduction (p. 2, line 59), suggesting that the Tg of the PLA is not far from room and close to the physiological temperature. A difference overcoming the dowble in both cases cannot be a "close" value. Please revise accordingly.

Round 2

Reviewer 2 Report

Corrections made are sufficient.

CO2 and H2O cannot be called nutrients.